# Evaluation of Treatment Outcome for Pneumonia among Pre-Vaccinated COVID-19 Patients with/without Comorbidity in a Public Hospital in Bengkulu, Indonesia

**DOI:** 10.3390/vaccines9121411

**Published:** 2021-11-30

**Authors:** Diana Laila Ramatillah, Siew Hua Gan, Syed Azhar Syed Sulaiman, Dama Puja, Usman Abubakar, Ammar Ali Saleh Jaber, Stefanus Lukas, Nina Jusnita

**Affiliations:** 1Faculty of Pharmacy, Universitas 17 Agustus 1945 Jakarta, Jakarta 14350, Indonesia; erlanggapuja23@gmail.com (D.P.); stefanus.lukas@uta45jakarta.ac.id (S.L.); nina.jusnita@uta45jakarta.ac.id (N.J.); 2School of Pharmacy, Monash University Malaysia, Jalan Lagoon Selatan, Bandar Sunway 47500, Selangor, Malaysia; gan.siewhua@monash.edu; 3Discipline of Clinical Pharmacy, School of Pharmaceutical Sciences, Universiti Sains Malaysia, Gelugor 11800, Penang, Malaysia; sazhar@usm.my (S.A.S.S.); pharmusman@usm.my (U.A.); 4Department of Clinical Pharmacy and Pharmacotherapeutics, Dubai Pharmacy College for Girls, Dubai 19099, United Arab Emirates; Dr.ammar@dpc.edu

**Keywords:** COVID-19, Pneumonia, green zone, pre-vaccination

## Abstract

Pneumonia is one of the common complications of SARS-CoV-2 infection where most patients have moderate to severe symptoms that pose a higher risk for death. This study aims to evaluate the treatment outcome of COVID-19-associated Pneumonia among patients with/without comorbidity in a public hospital in Indonesia. This is a retrospective cohort study involving unvaccinated confirmed COVID-19 patients admitted to the hospital between March and December 2020. All confirmed COVID-19 patients with Pneumonia (n = 1522) treated at the hospital were included. The majority of patients (99%) had mild COVID-19 symptoms while the remaining had moderate symptoms. The median age was about 32 years old and the average treatment duration was 6.25 ± 1.83 days. Most patients (88.8%) received a combination of azithromycin and oseltamivir. There was a very significant relationship (*p* < 0.001) between comorbidities with treatment and duration of treatment of Pneumonia in COVID-19 patients. Although most patients had Pneumonia and comorbidities, they were successfully treated with azithromycin and oseltamivir combination following approximately five days of treatment.

## 1. Introduction

Several coronaviruses cause respiratory tract infections in humans ranging from coughs and colds to more serious ones [1]. The symptoms experienced are usually mild and appear gradually and some infected individuals may not show any signs and still feel healthy [1]. WHO officially declared COVID-19 as a pandemic on Wednesday, 11 March 2020 and the cases have been increasing ever since. According to the WHO, a pandemic is the scale of the spread of a condition globally throughout the world, including in Indonesia [2,3].

The Indonesian Ministry of Health recorded an increase in positive cases of COVID-19 on 5 November 2021, bringing the total to 4,246,802. The Ministry of Health has examined 47,709,218 specimens, with a cure rate of 84.0%. The cases of death with confirmed COVID-19 increased to 143,500 cases [4].

To suppress the spread of COVID-19, the race for the development of vaccines and the search for new drugs started when the disease became an endemic by the end of 2019 and a pandemic since March 2020 [1]. Nevertheless, to date, there remains no specific pharmacological therapy for this virus [5] although the WHO has approved the use of some anti-viral agents including Oseltamivir, Favipirafir, Remdesir, Lopinavir, Ritonavir, Darunavir and Chloroquine. Additionally, some antibiotics have been approved for emergency use against bacterial infection for COVID-19 patients with Pneumonia as their secondary infection [6,7]. It is also noteworthy to state here that vaccination in Indonesia only began in January 2021, [8] and with its dense population, the country has a lot of catching up to do.

Bengkulu is one of the provinces in Indonesia located in Sumatera Island which has a lower COVID-19 mortality rate [4]. At the time of investigation, Bengkulu is classified under the green zone in Indonesia [4]. Although a lockdown was applied, the number of COVID-19 and deaths was included in the lower category, compared to other provinces during 2020 [4]. Therefore, in this area, patients can still be admitted to the hospital for COVID-19 treatment without having to queue [4]. In this study, we aim to evaluate the treatment outcome for Pneumonia among pre-vaccinated COVID-19 patients with/without comorbidity in a public hospital in Bengkulu, Indonesia.

## 2. Materials and Methods

### 2.1. Study Design and Setting

This is a retrospective cohort study involving 1522 laboratory-confirmed COVID-19 patients admitted to the green zone area at a public hospital, Bengkulu. All patients aged 17 years and above with confirmed diagnoses were included. Patients with cancer, HIV/AIDS, pregnancy, and lupus were excluded.

### 2.2. Ethical Approval

The research was approved by a local institutional ethics committee at the medical ethics committee from the polytechnic of health ministry Bengkulu with reference number: NO.KEPK/001/05/2021, which complies with the Declaration of Helsinki.

### 2.3. Data Collecting and Handling

Following ethical clearance, permission was requested from the hospital to access the medical records. Demographic and clinical data were screened and recorded based on patients’ medical records. Data analysis was conducted using Fisher Exact, Chi-Squared, Mann-Whitney and Kruskal Wallis. Due to the retrospective nature of the study, informed consents from the patients were not necessary. All data were kept anonymous and confidential. Subsequently, the data were entered into the Microsoft Excel 2013 program and were processed using an SPSS version 26.0 program.

### 2.4. Confirmation of Positive SARS-CoV-2 and Pneumonia

Before admission to the hospital, polymerase chain reaction (PCR) test results for SARS-CoV-2 from the nose/throat/airway aspirate swabs were confirmed to be positive. All patients were confirmed to have Pneumonia based on their chest X-rays on admission to the hospital (the first day of hospitalization right after the confirmation of a positive PCR test).

The type of Pneumonia that the patients have was community-acquired Pneumonia (CAP) which was acquired outside of a medical or an institutional setting. Pneumonia was confirmed based on the chest X-rays on the first day of admission following positive confirmation of COVID-19. As for the disease severity, these patients were categorized as either mild or moderate Pneumonia. In terms of disease severity, mild Pneumonia is indicated by a mild fever, dry cough, and flu symptoms, while moderate is described by cough and shortness of breath [9]. The staging was based on the information acquired from the medical record.

### 2.5. Clinical Outcome Parameters

In this study, recovery and death are two clinical outcome categories defined. The recovery parameter was based on the PCR test results for SARS-CoV-2 from either the nose/throat/airway aspirate swabs. The PCR findings must be negative at least two times in a row within 24 h. It is not compulsory to heal Pneumonia in totality although all patients were discharged in good condition. In fact, based on the COVID-19 treatment guideline in Indonesia, a negative PCR test is the only primary parameter for patient discharge [10].

Treatment outcome was based on the clinical evaluation, including the duration of treatment and hospitalization.

### 2.6. Antibiotic and Antiviral Selection

Patients with mild COVID-19 (n = 1507) were administered with azithromycin + oseltamivir, while those categorized as having moderate symptoms (n = 15) received ceftriaxone + favipiravir.

Based on the Indonesian COVID-19 treatment guideline, oseltamivir is used for flu-like symptoms (mild). Favipiravir was approved in Japan and China as an influenza treatment for moderate COVID-19 infection [6], and the said protocol was similarly applied in Indonesia.

## 3. Results

In this study, a total of 1522 patients were recruited with an average age of approximately 32 years. Azithromycin + oseltamivir was the most common treatment used. There was a very significant relationship between complication with treatment and duration of treatment (Table 1). All patients survived the disease.

The type of treatment (Azithromycin + Oseltamivir and Ceftriaxone + Favipiravir) and gender were associated with the Pneumonia type (in this case, either with or without complication) based on a Chi-Squared test. The age and duration of treatment (as continuous data) were correlated with the type of Pneumonia (either with or without complication) based on Mann-Whitney test.

Comorbidities were associated with the treatment type using a Fisher Exact test. A similar test (Fisher exact test) was also used to analyze the correlation between severity and comorbidities. The age and duration of treatment (as continuous data) were correlated with comorbidities using a Kruskal Wallis test.

Before deciding on the type of analysis used, a normality test was conducted. Based on the test, the age and duration of treatment were found not to be normally distributed and for this, Mann-Whitney and Kruskal Wallis were used.

Besides Pneumonia, approximately 9.4% of the patients had asthma, 4.6% had hypertension, 5.1% had tuberculosis, 2.7% had diabetes mellitus and 2.2% had uric acid (Table 2).

Most (99.0%) patients had mild symptoms, while only 1% had moderate symptoms. Additionally, only 8.6% of the patients had pre-existing comorbidities, with 99.2% having mild symptoms (Table 3). The average age of patients with asthma was approximately 27 years with duration of COVID-19 treatment of about five days. This was similarly seen in COVID-19 patients with DM. It was also noteworthy that COVID-19 patients with acute gouty attack and high uric acid as well as Pneumonia were approximately of a slightly higher age group (31 years old) and tended to receive a longer duration of treatment (15 days).

## 4. Discussion

To the best of our knowledge, this is the first study to evaluate the treatment outcome of COVID-19 in patients with Pneumonia in Indonesia. This study involves 1522 patients who were hospitalized due to COVID-19 in a public hospital. According to the WHO, COVID-19 tends to co-exist with Pneumonia, which began with a report from China to the World Health Organization (WHO) where 44 patients with severe Pneumonia were first identified in Wuhan City, Hubei Province, China, on the last day of 2019 (WHO, 2020) [1]. Since COVID-19 was announced as a pandemic, every country has applied the lockdown to curb its spread, including Indonesia [1]. Unfortunately, despite lockdowns, infection and mortality rate continue to increase in Indonesia, especially at the end of 2020 and in the middle of 2021 [8]. 

The Indonesian COVID-19 Handling Task Force segregated patients into green, yellow, red and black zone areas [8] based on the disease severity. Jakarta, being the capital and the largest city in Indonesia, remains in the red zone continuously and it was even designated as a black zone at the peak of the COVID-19 pandemic [8]. In contrast, Bengkulu wass marked as a green zone at the height of the pandemic in 2020. Based on a report published on the Bengkulu portal tracking for the rate of infection, the total positive cases of COVID-19 was 3552 as of 27 December 2020, with only 114 deaths [11]. Nevertheless, by 19 October 2021, the positive cases of COVID-19 in Bengkulu have risen to 23,083, with 404 patients succumbing to death by the infection [12] although it continues to be in the green zone category.

The median age of the COVID-19 patients in the current study was relatively low (32 years). It is plausible that the lower age group seen among COVID-19 patients in the current study is attributed to the fact that the younger age group in Bengkulu are the main breadwinner of their families while the elderly tend to stay at home. A previous study in Jakarta, Indonesia revealed a slightly higher age group with most patients (44.4%) between 39 and 58 years [13]. A study in China reported that most patients (47.7%) were aged between 9 and 58 years old [14] while in another study, the mean age of COVID-19 patients was stated at 50 years [15], and in the Western countries it was between 59 and 78 years old [16]. A study in Japan reported an even higher age range for COVID-19 patients (between 64 and 72 years) [17] which reflects its status as having “super-aged” society. 

A study in China reported that older patients (>50 years old) with severe dyspnea, chest pain and cough associated with Pneumonia tend to have worse clinical conditions [18]. In comparison, in our study, all patients had Pneumonia with only some having other comorbidities such as asthma, hypertension, tuberculosis, diabetes mellitus and uric acid, surviving perhaps due to their younger age brackets. Furthermore, most patients were in the mild disease categories; with only a single patient in the moderate category. This observation is important since early detection [19] and staging or severity of the disease [20] have been reported to contribute to the survival of COVID-19 patients.

In contrast, patients with severe or critical illness had a high mortality rate [20]. Based on the study by Pollan et al., most patients who came to the hospital had respiratory disease [21] because approximately a third of people with SARS-CoV-2 infection are asymptomatic and therefore, declined treatment. Similarly, Ting et al. reported that among COVID-19 patients, the most common comorbidities were hypertension, followed by diabetes [22]. This is consistent with reports that a history of cardiovascular disease and impaired renal function are related to COVID-19–associated mortality. It is plausible that patients with these comorbidities have more prominent target organ damage, which increases susceptibility to COVID-19 as well as risk of adverse outcomes. Similarly, another study reported that most COVID-19 patients had hypertension and diabetes with the median age of approximately 63 years [23]. Based on the study by Sanyaolu et al., COVID-19 patients with diabetes and hypertension aged above 65 years had higher severity of the illness and mortality [24] because patients with persistent hypertension and diabetes have significantly more marked endothelial damage and hence are more likely to develop a more severe course and disease progression.

Several studies reported a high mortality rate of COVID-19 patients and Pneumonia with multiple comorbid conditions [25,26]. In fact, a study in Mexico reported a higher mortality rate among those having COVID-19 and Pneumonia due to poorer response to treatment plans [27] or multi comorbid conditions [28]. Another reported that higher body mass index (BMI) with diabetes and hypertension are at great risk of death [26] since obese patients (BMI > 30 kg/m^2^) are already having an increased risk of mortality due to the type 2 diabetes.

In this study, there was a very significant relationship between comorbidities and duration of treatment. A retrospective study of middle-aged and elderly patients with COVID-19 reported that older people are more susceptible to the disease and are more likely to be admitted to the Intensive Care Unit (ICU) with a higher mortality rate [6,24]. Similarly, those with multi comorbid conditions have a longer duration of hospitalization [29] as they have a higher tendency to develop a more severe course and development of the disease. Nevertheless, this was not seen among our patients with a younger median age.

The most effective outcome for the combination of antibiotics with antiviral agents in the investigated cohort of patients was oseltamivir and azithromycin (for mild symptoms) while for moderate symptoms, ceftriaxone and favipiravir were useful. It is noteworthy to say that the said regimen was adopted before the recommendation by the WHO prior to the introduction of standard regimens against COVID-19 which exclude the use of oseltamivir [6,7]. Nevertheless, based on the report by Rosa et al., oseltamivir which is approved for influenza A and B [30] can act by inhibiting the viral neuraminidase, consequently blocking the release of viral particles from the host cells thus reducing the viral spread in the respiratory tract. Additionally, oseltamivir was used during the COVID-19 epidemic in China, with or without antibiotics and corticosteroids [30]. As protease inhibitors, the combination of lopinavir/ritonavir caused a reduction in viral loads and ameliorated viral symptoms during the treatment period by blocking the action and proliferation of protease enzymes (3CL^pro^), thereby disrupting the process of viral replication and release from host cells [31,32]. Additionally, ritonavir inhibits the metabolizing enzyme cytochrome P450 3A and thus elevates the half-life of lopinavir.

The antiviral treatments from human pathogenic CoVs include neuraminidase inhibitors like oral oseltamivir, which have been used in China hospitals for COVID-19 cases [31,33]. Nevertheless, no study has demonstrated the effectiveness of oseltamivir in the treatment of SARS-CoV-2 [31,34]. Based on the study by Kumar et al., the most effective drug as an antiviral for COVID-19 is lovinapir [35] while the clinical trial in Japan indicated that favipiravir showed good effectiveness for COVID-19 [36]. Favipiravir is a selective and potent broad spectrum RNA-dependent RNA polymerase (RdRp) inhibitor. Favipiravir is incorporated into the nascent viral RNA by error prone viral RdRp, which leads to chain termination and viral mutagenesis. Azithromycin and ceftriaxone are antibiotics utilized against COVID-19 [10] in Indonesia. 

In this study, the male gender tends to be more predominantly affected with COVID-19 as compared to females. Again, it is possible that this phenomenon is contributed by the fact that males are the main breadwinner in Indonesia and the fact that many Indonesian males are smokers (68.1%) [37] since smoking can impact both adaptive and innate immunity [38]. Similarly, other studies in Israel reported a higher incidence in males as compared to females [39] as well as in China (52.7%) [40] and Great Britain (57.75%) [41].

The average length of hospitalization of COVID-19 patients in this study was seven days. In comparison, COVID-19 patients with Pneumonia had a length of hospitalization of approximately six days while COVID-19 Pneumonia patients with comorbidity had a slightly longer duration of treatment of about eight days. Nevertheless, the study by Daniels et al. reported a shorter median time from hospitalization to severe disease (two days) while the median time to recovery was seven days [42]. Another study reported that elderly patients (>68 years) had to be treated for a longer duration than patients aged (38–58) because of their propensity to develop a more severe course and progression of SARS-CoV-2 infection due to the likely presence of comorbidities that come with ageing [43]. Therefore, it is essential to increase prevention efforts and to optimize health services [44] in patients with Pneumonia and/or comorbidities especially in those in the higher age group.

The study has some limitations where no data on non-steroidal anti-inflammatory drug (NSAID) use were available since the study has been conducted in a district hospital. The said data (if available) can provide a more consistent overview of the treatment outcome among the cohort of patients. Moreover, patient data retrieval is rather complicated in Indonesia since it is directly related to the quality of the data recording. Additionally, since the study was conducted before the introduction of vaccines, future studies are needed to compare the treatment outcomes based on the clinical outcome and duration of treatment following vaccine introduction to provide a more global picture of the situation.

## 5. Conclusions

All COVID-19 patients with Pneumonia survived with a treatment duration of approximately five days. A significant relationship was found between the duration of treatment and comorbidity. Therefore, great attention and time should be given to patients with comorbidities to avoid any further unfavourable outcomes, especially in those having moderate symptoms to prevent transitioning to more severe disease.

## Figures and Tables

**Table 1 vaccines-09-01411-t001:** Correlation between Pneumonia COVID-19 complication with and without comorbidity with the socio-demographics, treatment given and duration of treatment.

Variable	Frequency (%)			*p*-Value
	Overall (n = 1522)	Pneumonia in COVID-19 Patients with Complication and without Comorbidity (n = 1391)	Pneumonia in COVID-19 Patients Complication and with Comorbidity (n = 131)	
**Median of Age**	32	32	31	0.904 ^#^
**Gender**				0.743 *
Male	1322 (86.9)	1207 (86.8)	115 (87.8)	
Female	200 (13.1)	184 (13.2)	16 (12.2)	
**Treatment**				0.000 *
Azithromycin + Oseltamivir	1351 (88.8)	1248 (89.7)	103 (78.6)	
Ceftriaxone + Favipiravir	171 (11.2)	143 (10.3)	28 (21.4)	
**Median duration of treatment (days)**	**5**	**5**	**7**	**0.000 ^#^**

* Chi-Squared test; ^#^ Mann-Whitney.

**Table 2 vaccines-09-01411-t002:** Correlation between treatment type and comorbidity.

Comorbidities/Complication	Treatment Type n (%)		*p*-Value *
	Azithromycin and Oseltamivir	Ceftriaxone and Favipiravir	
Pneumonia	1248 (93.4)	143 (83.6)	0.002
Pneumonia + Asthma	41 (3.0)	11 (6.4)	
Pneumonia + Hypertension	23 (1.7)	5 (2.9)	
Pneumonia + Tuberculosis (TB)	13 (1.0)	7 (4.1)	
Pneumonia + Diabetes Mellitus (DM)	12 (0.9)	3 (1.8)	
Pneumonia + Uric Acid (UA)	14 (1.0)	2 (1.2)	
Total	1351	171	

* Fisher Exact Test.

**Table 3 vaccines-09-01411-t003:** Correlation between comorbidities/complications and severity.

Variable	Comorbidities/Complication N (%)						*p*-Value
	P	P + A	P + H	P + TB	P + DM	P + UA	
Severity							0.742 *
Mild	1377 (99.0)	51 (98.1)	28 (100.0)	20 (100.0)	15 (100.0)	16 (100.0)	
Moderate	14 (1.0)	1 (1.9)	0	0	0	0	
Median of Age	32	27.5	32.5	33.5	27	31.5	0.665 ^#^
Total	32						
Median duration of treatment (days)	5	5	7	7	5	15.5	0.000 ^#^
Total	5						

* Fisher Exact Test, ^#^ Kruskal Wallis. Note: P = Pneumonia; P + A = Pneumonia + Asthma; P + H = Pneumonia + Hypertension; P + TB = Pneumonia + Tuberculosis; P + DM = Pneumonia + Diabetes Mellitus; P + GA = Pneumonia + Gouty Attack.

## Data Availability

https://figshare.com/blog/figshare_as_a_data_repository/410 (accessed on 13 October 2021).

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
