# Peer review of "Evaluation of Treatment Outcome for Pneumonia among Pre-Vaccinated COVID-19 Patients with/without Comorbidity in a Public Hospital in Bengkulu, Indonesia"

_vaccines, 2021, doi:10.3390/vaccines9121411_

Round 1
Reviewer 1 Report
Dear Authors
The manuscript evaluates the treatment outcome of COVID-19 pneumonia among patients with/without comorbidity in a public hospital in Indonesia. A retrospective cohort study involving unvaccinated confirmed Covid-19 patients were performed. In this study, all confirmed COVID-19 patients with pneumonia treated at the hospital were included. Most patients received a combination of azithromycin and oseltamivir. This study found a very significant relationship between comorbidities with treatment and duration of treatment of Pneumonia in COVID-19 patients.
The study from a scientific point of view seems to be well done and presents good results, from where to derive valid conclusions.
Abstract includes introductory statement that outlines the background and significance of the study.
Introduction summarizes relevant research to provide context and clearly state the problem. The topics are well developed and confronted to other publications.
The research methods used ensure the reliability of the obtained results.
The results are presented clearly. In Table 1 and Table 2 coorect the legend “Correlation” with “association”.
The discussion section interprets the findings in view of the results obtained in this and in past studies on this topic.
References cited are recent and have a high relevance to the problem.
Author Response
Answer : Thank you for the comment. We have now changed the term to “Antibiotic and Antiviral Selection” to be accurate. Thank you for the valuable comment.
Reviewer 2 Report
Manuscript ID (Vaccines-1473988) entitled “Evaluation of Treatment Outcome for Pneumonia among pre-Vaccinated COVID-19 patients with/without comorbidity in a public Hospital in Bengkulu, Indonesia”
Significance:
Ramatillah et al. reported clinical outcomes of Pneumonia treatment among 1522 unvaccinated COVID-19 patients with the median age of 32-years-old in a public hospital in the green zone of Indonesia for March to December 2020. Clinical study results in this age group are limited hence it is valuable data to understand the age-associated clinical symptoms, frequency of comorbidity among COVID 19 patients, and effective therapeutic treatments.
Weaknesses/limitations
The description of mild and moderate symptoms is required, and the degree of severity needs to be defined.
It was presumptuous that pneumonia was deemed to be caused by COVID-19, excluding the possibility that co-infections or secondary infections with other bacteria could be associated with pneumonia. Although the treatments included combinatory therapies of an antibiotic and an anti-viral reagent, the parameter of recovery was limited to the negative PCR result for SARS CoV-2 excluding parameters to evaluate the recovery from pneumonia. More measurable parameters evaluating the severity of pneumonia in the course of the therapeutic drug treatments should have been included.
The majority (91.4%, 1391/1522) of the COVID 19 patients with pneumonia only and 8.6 % (131/1522) of the patients with comorbidity were treated with either (Azithromycin and Oseltamivir) or (Ceftriaxone and Favipiravir). Overall, it is unclear how the authors achieved a significant relationship between the duration of the treatment and comorbidity under the study conditions in which there were insufficient cohort sample sizes to compare the frequency of comorbidity with different severity of clinical symptoms and different therapies.
The major weakness of the manuscript is the lack of a description of how statistical analyses were performed. Please describe variables compared such as symptoms, comorbidities, duration of therapy including the description of any assumptions of corrections, such as tests of normality and adjustment for multiple comparisons.
Please read following specific comments on the main body of the manuscript.
Page 2, line 70, the description of clinical symptoms of the participating patients and the severity of pneumonia are required.
Page 2, Materials and Methods should include a section for Statistical analyses: Please describe variables compared such as symptoms, comorbidities, duration of therapy including the description of any assumptions of corrections, such as tests of normality and adjustment for multiple comparisons.
Page 3, Table 1. The overall numbers of males and females are incorrect.
The total number of males is 1322 (1207 + 115 = 1322) instead of 1391
Also, the total number of females is 200 (184+ 16 =200) instead of 131. Please clarify.
Page 4, Table 3, it is unclear what variables were compared and how p-values were calculated.
Linked to Table 3, Page 5 lines 181-182, the statement ‘In this study, there was a very significant relationship between comorbidities and duration of treatment.” needs clarifications
- Does this mean in comparison to the 5 days duration of treatment in 1377 pneumonia only patients against the 7 days duration of treatment in 28 (P+H) patients and 20 (P+TB) patients? Again, it is not clear how the authors achieved a statistical significance. The description of how statistical analyses were performed is required including any assumptions of corrections, such as tests of normality, categorized variables, and adjustment for multiple comparisons and others applicable.
Page 5, Lines 185-187: In this study, age is unlikely a factor affecting the outcome of therapeutic effects of drug treatments as the study subjects seemed to be within the same age bracket. The current study did not compare data from elderly COVID-19 patients with mild/moderate pneumonia with or without comorbidity side-by-side.
- Be cautious in the use of “the duration of treatment” and “the duration of hospitalization”
- Although it may be plausible that the duration of treatment might be longer in elderly population, the interpretation of the current study result by the author is inappropriate.
Page 5, lines 189-191 need citations.
Overall, the Discussion section would have been more focused to related to the finding(s).
I think that the title of each of the tables does not represent the information presented in Table 1 and 2 in the current version of the manuscript. It is unclear to me what is compared with what to show correlative relationship. These need to be revised - maybe it makes a better sense if Tables 1 and 2 are combined together to show socio-demography, the spectrum of comorbidity of the Pneumonia COVID-19 patients. The title can be Table 1. Comorbidity of the confirmed COVID 19 test-positive patients with pneumonia.
Then, show the result - the correlation with 95% confidence intervals- in a separate table or figure.
Author Response
- The description of mild and moderate symptoms is required, and the degree of severity needs to be defined.
Answer : Thank you for the valuable comment. Mild pneumonia is indicated by a mild fever, dry cough, and flu symptoms, while moderate is described by cough and shortness of breath [51]. The said information has now been added to the newly-edited manuscript.
- More measurable parameters evaluating the severity of pneumonia in the course of the therapeutic drug treatments should have been included.
Answer: We thank the reviewer for another point of view. However, the case management may vary from country to country where locally, it is not compulsory to heal pneumonia in totality although all patients were discharged in good condition. In fact, based on the Covid-19 treatment guideline in Indonesia, a negative PCR test is the only primary parameter for patient discharge. Additionally, patient data retrieval is a challenge in Indonesia since it is directly related to the quality of the data recording. We have included the said information in the study limitation. We thank the reviewer for the comment, nevertheless.
- Most COVID 19 patients (91.4%, 1391/1522) were with only pneumonia. Approximately 8.6% (131/1522) of the patients with comorbidities were treated with either 1) azithromycin and oseltamivir combination or 2) ceftriaxone and favipiravir. Overall, it is unclear how the authors achieved a significant relationship between the duration of the treatment and comorbidity under the study conditions in which there were insufficient cohort sample sizes to compare the frequency of comorbidity with different severity of clinical symptoms and different therapies.
Answer: Thank you for the valuable comment. The reviewer has a point. Nevertheless, we conducted the analysis using a Fisher Exact Test for the insufficient variable number whenever that occurs. Thank you.
- The major weakness of the manuscript is the lack of a description of how statistical analyses were performed. Please describe variables compared such as symptoms, comorbidities, duration of therapy including the description of any assumptions of corrections, such as tests of normality and adjustment for multiple comparisons.
Answer : We appreciate the reviewer’s attention to detail in terms of statistical test. To be precise, the type of treatment (Azithromycin + Oseltamivir and Ceftriaxone + Favipiravir) and gender were associated with the type of pneumonia (either with or without complication) based on a Chi-Squared test. Age and duration of treatment (as continuous data) were correlated with the type of pneumonia (either with or without complication) based on Mann-Whitney test.
Comorbidities were associated with the treatment type using a Fisher Exact test. A similar test (Fisher exact test) was also used to analyze the correlation between severity and comorbidities. The age and duration of treatment (as continuous data) were correlated with comorbidities using the Kruskal Wallis test.
Before deciding the type of analysis used, the normality test was conducted. Based on the test, the age and duration of treatment were found not to be normally distributed and for this, Mann-Whitney and Kruskal Wallis were used. The said information has now been added to the newly-added manuscript.
- Page 3, Table 1. The overall numbers of males and females are incorrect.
The total number of males is 1322 (1207 + 115 = 1322) instead of 1391
Also, the total number of females is 200 (184+ 16 =200) instead of 131. Please clarify.
Answer : Apology, indeed, we made a mistake in the calculation of the total samples (male and female). We have corrected the said information in the table. Thank you for pointing this out.
- Page 4, Table 3, it is unclear what variables were compared and how p-values were calculated.
Answer : We have corrected the said information as in the table (severity= variable and below that the severity group). Thank you very much.
- Linked to Table 3, Page 5 lines 181-182, the statement `In this study, there was a very significant relationship between comorbidities and duration of treatment.” needs clarifications
Does this mean in comparison to the 5 days duration of treatment in 1377 pneumonia only patients against the 7 days duration of treatment in 28 (P+H) patients and 20 (P+TB) patients? Again, it is not clear how the authors achieved statistical significance. The description of how statistical analyses were performed is required including any assumptions of corrections, such as tests of normality, categorized variables, and adjustment for multiple comparisons and others applicable.
Answer : The reviewer has a point in terms of clarity of our write-up. Actually, the duration of treatment (as continuous data) were correlated with comorbidities using the Kruskal Wallis test.
Before deciding the type of analysis used, the normality test was compulsory to be checked. From the test, it can be known that age and duration of treatment were not a normal distribution, and for this, Mann-Whitney and Kruskal Wallis tests were used.
We realise that the confusion would not have arisen had the matter been made clear in the beginning (as in point no 4 above). The said information has now been included.
- Page 5, Lines 185-187: In this study, age is unlikely a factor affecting the outcome of therapeutic effects of drug treatments as the study subjects seemed to be within the same age bracket. The current study did not compare data from elderly COVID-19 patients with mild/moderate pneumonia with or without comorbidity side-by-side.
- Be cautious in the use of “the duration of treatment” and “the duration of hospitalization”
- Although it may be plausible that the duration of treatment might be longer in elderly population, the interpretation of the current study result by the author is inappropriate.
Answer : Indeed, the reviewer has a point that age is an unlikely factor affecting the outcome of therapeutic effects of drug treatments as the study subjects seemed to be within the same age bracket.
Nevertheless, we have used Mann-Whitney/ Kruskal Wallis tests. Therefore, the age becomes a continuous data to be analyzed against the type of pneumonia or comorbidities. We thank the reviewer for this comment, nevertheless.
We are cautious about using “the duration of treatment” and “the duration of hospitalization” interchangeably in the manuscript even though they may not mean the same thing. However, we decided to utilise “the duration of treatment” since in this study, all patients were discharged with a negative PCR test result.
I think that the title of each of the tables does not represent the information presented in Table 1 and 2 in the current version of the manuscript. It is unclear to me what is compared with how to show correlative relationships. These need to be revised - maybe it makes a better sense if Tables 1 and 2 are combined together to show socio-demography, the spectrum of comorbidity of the Pneumonia COVID-19 patients. The title can be Table 1. Comorbidity of the confirmed COVID 19 test-positive patients with pneumonia.
Then, show the result - the correlation with 95% confidence intervals- in a separate table or figure.
Answer : We thank the reviewer for the comment. We have attempted to combine the two tables (Table 1 & 2) accordingly. However, we have some challenges since the variables are different. We hope that the reviewer is ok with our data presentation.
Reviewer 3 Report
Evaluation of Treatment Outcome for Pneumonia among Pre-Vaccinated COVID-19 patients with/without Comorbidity in a Public Hospital in Bengkulu, Indonesia
The researcher aimed to evaluate the treatment outcome for pneumonia among pre-vaccinated COVID-19 patients with/with- out comorbidity in a public hospital in Bengkulu, Indonesia
I reckon the topic is highly relevant with the COVID-19 pandemic. However, there are some major fundamental issues with this manuscript that should be addressed.
- Type of Pneumonia is not specified
For examples:
Hospital-acquired pneumonia (HAP). This type of bacterial pneumonia is acquired during a hospital stay. It can be more serious than other types, as the bacteria involved may be more resistant to antibiotics.
Community-acquired pneumonia (CAP). This refers to pneumonia that’s acquired outside of a medical or institutional setting.
Ventilator-associated pneumonia (VAP). When people who are using a ventilator get pneumonia, it’s called VAP.
Aspiration pneumonia. Inhaling bacteria into your lungs from food, drink, or saliva can cause aspiration pneumonia. It’s more likely to occur if you have a swallowing problem, or if you’re too sedated from the use of medications, alcohol, or other drugs.
- Diagnosis of pneumonia is not confirmed objectively
- Diagnosis should be confirmed by chest radiography or ultrasonography.
Other minor comments:
2.6. Antibiotic Selection
Patients with mild COVID-19 (n = 1507) were administered with azithromycin + oseltamivir, while those categorized as having moderate symptoms (n = 15) received ceftriaxone + favipiravir
Comments: The term Drug should be used instead of Antibiotic
Author Response
Responses to Reviewer 3’s comments
Answer : Prior to hospital admission, polymerase chain reaction (PCR) test results for COVID-19 from the nose/throat/airway aspirate swabs were confirmed to be positive. All patients were confirmed to have Pneumonia based on their chest X-rays on admission to the hospital. The type of pneumonia that the patients have were Community-Acquired Pneumonia (CAP). The said information has now been added to the newly-added manuscript.
The reviewer is sharp to point this out. Pneumonia was confirmed based on the chest X-rays on the first day of admission following positive confirmation of Covid-19. As for the disease severity, these patients were categorized as either mild or moderate Pneumonia.
Similar to our responses to reviewer 2, a mild Pneumonia Covid-19 is shown as having a mild fever, dry cough and flu symptoms, while a moderate pneumonia is described as having cough and shortness of breath [51]. We thank the reviewer for the comments.
2.6. Antibiotic Selection
Patients with mild COVID-19 (n = 1507) were administered with azithromycin + oseltamivir, while those categorized as having moderate symptoms (n = 15) received ceftriaxone + favipiravir
Comments: The term Drug should be used instead of Antibiotic
Responses to Reviewer 3’s comments
Answer : Thank you for the comment. We have now changed the term to “Antibiotic and Antiviral Selection” to be accurate. Thank you for the valuable comment.
Reviewer 4 Report
This paper reports a statistical evaluation of treatment outcomes of COVID-19 pneumonia among patients with/without comorbidity in a public hospital in Indonesia. This data are in the form of a retrospective cohort study involving unvaccinated confirmed Covid-19 patients admitted to the hospital.
On a first reading, the statistical data and analysis appear to be competently done and the paper is reasonably well written. Here is what to do in a revision: Add carefully constructed paragraphs to Section 3 Results that describe in detail the contents of Tables 1, 2, and 3, their substantive meaning, how the correlations are estimated and their interpretation and how the statistical significance of the relationship in the tables are measured, estimated and assessed for statistical significance. This will be very important to making the findings of the paper meaningful, interpretable, and assessible to readers of the paper.
Author Response
Add carefully constructed paragraphs to Section 3 Results that describe in detail the contents of Tables 1, 2, and 3, their substantive meaning, how the correlations are estimated and their interpretation and how the statistical significance of the relationship in the tables are measured, estimated and assessed for statistical significance. This will be very important to making the findings of the paper meaningful, interpretable, and accessible to readers of the paper.
Responses to Reviewer 4’s comments
Answer
Similar to our Responses to reviewer 2, the type of treatment (Azithromycin + Oseltamivir and Ceftriaxone + Favipiravir) and gender were associated with the type of pneumonia (either with or without complication) based on a Chi-Squared test. Age and duration of treatment (as continuous data) were correlated with the type of pneumonia (either with or without complication) based on Mann-Whitney test.
Comorbidities were associated with the treatment type using a Fisher Exact test. A similar test (Fisher exact test) was also used to analyze the correlation between severity and comorbidities. The age and duration of treatment (as continuous data) were correlated with comorbidities using the Kruskal Wallis test.
Before deciding the type of analysis used, the normality test was conducted. Based on the test, the age and duration of treatment were found not to be normally distributed and for this, Mann-Whitney and Kruskal Wallis were used. The said information has now been added to the newly-added manuscript. We appreciate the reviewer’s attention to detail in terms of statistical tests. Thank you.
Round 2
Reviewer 2 Report
I thank the authors for revising the manuscript.
There is a few minor points required for revision, mainly due to the incorrect use of terminology-"COVID-19" and "SARS-CoV-2"-and/or inconsistent use of terminology -"Pneumonia" vs. "pneumonia".
The authors inter-changeably used the name of the disease coronavirus disease 2019 (COVID-19) and the causative virus “SARS-CoV-2” throughout the manuscript. There are several places that the terminology is used incorrectly. A few examples are as following:
Page 1 Abstract line 1, “….complication of COVID-19 infection” change from “COVID-19” to “SARS-CoV-2”
Page 2 section 2.4 subtitle, “Confirmation of positive Covid-19 and Pneumonia” should be corrected to ‘…positive SARS-CoV2…’
- Section 2.4, line 2, correct from “for COVID-19 from the nose/…” to “for SARS-CoV-2 from the nose”
- Section 2.5, line 2, correct from “the PCR test result for COVID-19…” to “…for SARS-CoV-2 the nose”
Page 7 line 11 “COVID-19 infection” should be “SARS-CoV-2 infection”
Page 1 Abstract line 3, correct from “… outcome of COVID-19 Pneumonia” to “COVID-19 associated Pneumonia”
Be consistent in the use of “COVID-19”- all upper case- instead of “Covid-19” or “covid-19” throughout the manuscript. Only few are indicated here as examples.
- Page 1, Abstract line 7, change from “Covid-19” to “COVID-19”
- Page 3 line 1, change from “Covid-19” to “COVID-19”
There are many “pneumonia” and “Pneumonia” with uppercase ‘P’ throughout the main body throughout the manuscript. Be consistent or define if ‘pneumonia’ and ‘Pneumonia’ is used for different designation. Only few are indicated here as examples.
- Section 2.4, line 5, two at the beginning and at the end of the sentence
- Section 2.4, line 9,
- Section 2.5, the last line of Page 2,
- Page 7, line 5
Author Response
Answer : Thank you for the useful comments. We have revised all according to the suggestion by the reviewer.
Reviewer 3 Report
The comments have not been well addressed as the methodology pitfalls have not been rectified.
Author Response
Answers :
Before admission to the hospital, polymerase chain reaction (PCR) test results for SARS-CoV-2 from the nose/throat/airway aspirate swabs were confirmed to be positive. All patients were confirmed to have Pneumonia based on their chest X-rays on admission to the hospital (the first day of hospitalization following a positive PCR test confirmation).
The type of Pneumonia that the patients have was community-acquired Pneumonia (CAP) which is acquired outside of a medical or institutional setting. Pneumonia was confirmed based on the chest X-rays on the first day of admission following positive confirmation of COVID-19. As for the disease severity, these patients were categorized as either mild or moderate Pneumonia. In terms of disease severity, mild Pneumonia is indicated by a mild fever, dry cough, and flu symptoms, while moderate is described by cough and shortness of breath [51]. The staging was based on the information in the medical record.
Reviewer 4 Report
This paper has been revised in response to the previous review. The revisions are good. The one thing that could be added at this point is references to conventional, accessible statistical methods publications for each of the statistical measures of association and significance cited in the text (Chi-squared, Fisher exact, etc.). This will be useful to readers who may not be statisticians.
Author Response
Answer : Thank you for the comments that our revisions are good. We have also revised the statistical parts accordingly as well.
Round 3
Reviewer 3 Report
I found the following explanation not satisfactory and did not address the methodological pitfalls and oversight
The following replies do not match with the raised concerns:
Before admission to the hospital, polymerase chain reaction (PCR) test results for SARS-CoV-2 from the nose/throat/airway aspirate swabs were confirmed to be positive. All patients were confirmed to have Pneumonia based on their chest X-rays on admission to the hospital (the first day of hospitalization following a positive PCR test confirmation).
The type of Pneumonia that the patients have was community-acquired Pneumonia (CAP) which is acquired outside of a medical or institutional setting. Pneumonia was confirmed based on the chest X-rays on the first day of admission following positive confirmation of COVID-19. As for the disease severity, these patients were categorized as either mild or moderate Pneumonia. In terms of disease severity, mild Pneumonia is indicated by a mild fever, dry cough, and flu symptoms, while moderate is described by cough and shortness of breath [51]. The staging was based on the information in the medical record.
Author Response
In this third revision, we have tried our best to address any methodological pitfalls and oversight as pointed by reviewer 3. While we are wary that the approach and diagnosis for pneumonia may vary from one locality to another, for a third world country like Indonesia, we are also aware that it is less than ideal. The below is the best information we can provide so far which is as close to the real scenario as possible. We hope that reviewer 3 is good with the suggestion:
Before admission to the hospital, polymerase chain reaction (PCR) test results for SARS-CoV-2 from the nose/throat/airway aspirate swabs were confirmed to be positive. All patients were confirmed to have pneumonia based on their chest X-rays on admission to the hospital (the first day of hospitalization following a positive PCR test confirmation).
The type of pneumonia that the patients have was community-acquired Pneumonia (CAP) which was acquired outside of a medical or an institutional setting. Pneumonia was confirmed based on the chest X-rays on the first day of admission following positive confirmation of COVID-19. As for the disease severity, the patients were categorized as either mild or moderate pneumonia. In terms of disease severity, mild pneumonia is indicated by mild fever, dry cough and flu symptoms, while moderate pneumonia is described by cough and shortness of breath [51]. The staging was based on the information acquired from the medical record.